# Body Awareness Does Not Need a Pedigree: Mixed-Breed Dogs Rely More on Self-Representation Than Social Learning in a Spatial Task

**DOI:** 10.3390/ani15030432

**Published:** 2025-02-04

**Authors:** Petra Dobos, Péter Pongrácz

**Affiliations:** Department of Ethology, ELTE Eötvös Loránd University, Pázmány Péter Sétány 1/c, 1117 Budapest, Hungary; dobospetra1@gmail.com

**Keywords:** body awareness, social learning, mixed-breed dogs, biological relevance

## Abstract

In natural circumstances, dogs usually have multiple options to solve a problem. Here, we tested companion dogs in a scenario where their reward was placed behind a transparent obstacle. The dogs had to find a way behind this obstacle, either by making a detour or using a shortcut (across an opening). The size of the opening was either comfortably large or just passable for the dogs. The dogs could also observe the experimenter demonstrating how to detour around the fence. The dogs had the chance to rely on their body-size awareness to decide whether they would opt for the large or the small opening, or for the demonstrated detour instead of using the shortcut. We found that dogs used their own size as a reference for deciding which opening would fit them better. Dogs who were highly interested in the demonstrator’s behavior were more likely to make a detour, even if the large opening was available for them. Observational learning and body awareness were tested for the first time together in a biologically relevant scenario. We only used mixed-breed dogs at this time, which provides good insights into dog behavior without the influence of task-related selection typical in purebred dogs.

## 1. Introduction

The adaptive nature of complex cognitive capacities in humans is rarely questioned, a fact that is further signified by the abundance of research that is connected to the various malfunctioning of these mental dimensions (e.g., language acquisition [1]; self-representation [2]; theory of mind [3]). However, the higher-order complex cognitive features of nonhuman species first must find their place within the biologically relevant framework of ethology and behavioral ecology, if we want to study them without the constant and often arbitrary referencing to human cognition [4]. If we adopt a viewpoint that favors evolutionary and ecological considerations, we realize that the various components of cognition, such as self-representation, have been evolving under the forces of selection, typical to the history of that species [5,6].

Dogs (*Canis familiaris*) are an excellent candidate for the ecologically valid investigation of nonhuman cognition. Since their domestication, forces of selection acted towards the dogs’ successful adaptation to the anthropogenic niche [7]. This process resulted in a species that shows strong dependence not only on human resources [8] but manifests sociocognitive capacities that also provide a dependence-based behavioral toolkit for smooth interactions with humans (e.g., dog–human attachment [9]; cooperation [10]; communication [11]; and social learning [12]). When ethologists investigate the sociocognitive capacities of dogs in scenarios that include interaction with humans and/or anthropogenic factors, it is feasible to formulate biologically relevant, functional hypotheses about the expected behavioral responses of dogs [6].

However, researchers investigating the capacity of self-representation in animals still prefer to focus on the ‘higher end’ (most complex) forms of self-recognition that develop in humans towards the later stages of their cognitive development (i.e., self-recognition in a mirror [13,14]). From an ethological perspective, the main problem with the top-down approach is the lack of ecological validity, in other words, by searching for higher-end cognitive attributes in non-human species, we can rarely assess the function (adaptivity) of this feature in the given species [15]. According to the modular framework of self-representation [16], the presence and relative importance of any module depends on the specific adaptation processes of that species. The modules range from the simplest forms of sensorimotor interference that provide the necessary feedback for differentiating the self from the environment [17] to more complex cognitive features such as episodic-like memory [18], or the formation of theory of mind [19]. Although domesticated species, especially dogs and cats (e.g., [20,21]), have recently been at the forefront of investigation of non-human sociocognitive capacities, apart from a few exceptions (e.g., [22,23]) they are seldom included in experiments about self-representation.

Body awareness is regarded as one of the fundamental building blocks of self-representation [4]. During navigation in the physical environment, having information about the body’s extension and other physical features has definite advantages and it is thought to be the phylogenetically earliest appearance of any kind of self-knowledge [24]. Dogs belong to a group of fast moving, pursuit-type predators, where collisions or becoming stuck in too narrow openings could have serious consequences. Their body size has been found to play an important role during social interactions, such as in agonistic encounters (e.g., [25,26]); therefore, a well-working, body-size awareness-based referencing system, would be highly adaptive for the dog.

The first evidence about body-size awareness in dogs came from a study in which a series of incrementally decreasing size, door-choice tasks found that dogs approached ‘too small’ openings with significantly longer latencies than ‘large enough’ openings [16]. These results suggest that dogs have an expectation of whether they would be able to pass through an opening or not. However, dogs only had one option to negotiate the obstacle: a single opening in a wall, which the subjects had to approach regardless of whether it was big enough for them to go through or not. In more naturalistic scenarios, animals often face several solutions to the same problem. Therefore, in another study, we tested whether dogs would rely on knowledge of their own size when facing a problem with multiple solutions, in which only one of these required body awareness. We offered a spatial problem to the dogs, where they could reach the reward by either making a detour around a transparent fence or making a shortcut through an opening that was either too small, or comfortably large, compared to the dog’s size. We found that if the door was large enough, dogs hardly ever made a detour, while they rarely (and then slowly) approached the too small openings and rather opted for a detour [27].

Apart from a recent experiment of Kobayashi et al. [28] on cleaning fish, body-size representation has always been tested in non-social circumstances. Thus, none of the methods mentioned so far took into consideration the highly social nature of dogs. Dogs rely not only on the behavior of conspecifics in problem-solving situations [29,30], but they also easily learn from the observation of human behavioral cues and templates (e.g., [31]). Learning from the observation of human demonstrators can sometimes even lead to suboptimal choices in terms of success [32,33], and the speed of problem-solving [34]. This, in theory, can be explained with the ultimately adaptive outcome of dogs’ dependency on humans [35], and their willingness to comply with the social rules of the anthropogenic environment [36]. Ecologically valid experiments should include the complex assortment of factors with already known effects to see how these interplay in a more realistic situation, therefore we re-designed our previous experiment [27] and included a component where dogs can observe a human demonstrator performing the less optimal (detour) solution to the task. In this way, dogs would be offered an optimal shortcut against a longer detour solution; however, the latter would be reinforced by repeated action from the human demonstrator [34]. At the same time, the suitability of the shortcut (opening size) could be assessed if the dogs relied on their body awareness [27].

There are abundant indications that the performance of dogs in sociocognitive problem-solving tasks is influenced by various extrinsic (e.g., training and housing [37]) and intrinsic factors (e.g., age [38]; head shape [39]; breed [6]). It was found that dogs’ decision-making regarding a detour or using the shortcut was associated with their cephalic index [27], namely brachycephalic dogs preferred to choose the opening ahead of them, while longer-headed dogs opted more for the detour. In our present experiment we decided to use only normocephalic dogs to avoid this confounder. Regarding dog breeds, their potential effect on body-size awareness has not been tested yet; however, there are ample indications regarding the associations between functional breed selection and dogs’ performance in social learning tasks. It was found that breeds of working dogs that were selected for cooperativity with humans show an especially high dependency on human communication and demonstrated behaviors; meanwhile, independent working dog breeds mostly disregard human communication and behavioral templates [40,41]. In our present paper, we wanted to test dogs’ decision-making based on social learning and body size awareness, without the confounding effect of breed selection, thus we assembled our sample from mixed-breed dogs only. Although, thus far, there are very few investigations that specifically targeted mixed-breed companion dogs (for an exception see [39]), we assumed that dogs that are not direct descendants of purebreds will show a balanced performance without either exceptionally high, or low dependence on human behavior.

This experiment included three conditions. We tested whether dogs would keep relying on an easy shortcut through a comfortably large opening if they had, or did not have, an a priori opportunity to observe a demonstrator who performed detours around the obstacle (‘large door without detour demo’ vs. large door with detour demo’). Reliance on body awareness was tested by using another condition, in which we offered the dogs a still passable, but less comfortable (‘small’) opening (large door without detour demo’ vs. small door without detour demo’). We had two hypotheses. According to the first, dogs will mostly opt for the shortcut over the longer detour, and they will rely on their body awareness to assess the opening size. In this case, we predicted that regardless of the detour demonstration, dogs will always prefer the large door, but in the case of the small door, the frequency of detours will increase. According to the second hypothesis, the social learning of the detour could override the preference for the shortcut. In this case, when dogs observed the demonstrator’s action, they would choose the large door less often than without demonstration. Body awareness would still have an effect in the case without detour demonstration groups, as dogs would prefer detours only in the case of the small door. Figure 1 shows a flow chart of the targeted phenomena, experimental design, and the main behavioral variables of our study.

## 2. Materials and Methods

### 2.1. Subjects

We tested N = 57 mixed-breed dogs that were more than 1 year old (Table 1), regardless of their sex (mean of age ± SD = 6.1 ± 3.6 year). Following the method of Gácsi et al. [39], we considered non-pedigreed dogs as being a ‘mixed breed’ if, according to the owner, (1) it was not a crossbreed of known purebred parents; and (2) it did not show an apparent visual match with any dog breeds known to us. Another criterion was that the dogs’ height at the withers had to be between 30 and 90 cm. Owners were recruited through social media platforms. We tested only normocephalic dogs, i.e., we did not recruit any dogs with strong brachycephalic features. We recorded the keeping conditions (indoor only; indoor–outdoor; outdoor only), and the training level of the dogs (none; training at home; dog school course; regular dog school; individual trainer; special sport/work training).

#### Exclusions

We excluded N = 12 dogs from the statistical analysis. A total of 11 dogs could not detour the fence during the closed-door trials, and N = 1 dog went between the experimental fence and the property border. Thus, the data from N = 45 subjects were entered into the statistical analysis.

### 2.2. Equipment

The dogs were tested at the Eötvös Loránd University’s campus on an outdoor grassy area. We used a 1 m high and 3 m long fence as an experimental device, which was constructed from transparent wire mesh, with a hole diameter of 20 mm, attached to a steel frame (Figure 2). One end of the device was fastened to the chain link fence on the property border, giving it an angle of 90°. The property border fence was not possible to detour around. The frame of the experimental fence prevented the dogs from digging under it. The tests were recorded with two cameras (Blow Go Pro4U (Garwolin, Poland) and Panasonic (Kadoma, Japan)) on tripods.

In the middle of the first 1 m wide section of the fence (i.e., the closest to the property border) we added an adjustable opening that could be set to one of five possible sizes (which we will call ‘doors’ for simplicity). Size adjustments were possible with the insertion of a series of opening reducers into the original (largest) opening. The largest opening had an upwards swinging door, which was secured in the upper (open) or lower position (‘closed door’ condition).

Door size was a fundamental detail of the experiment. The test groups were formed based on whether the dogs were provided with a ‘large’ or a ‘small but still suitable’ sized door. Depending on the testing group, each dog’s height was paired with a corresponding ‘small’ or ‘large’ door-size (Table 2). Crucially, in this study we intended to provide the dogs with a passable shortcut option even in the case of the ‘small door’. Based on the results of [16,27], although dogs noticed and approached very small openings (only one-third as high as the dog’s height at the withers), eventually they did not go through these. For our present experiment, such short doors would be too small. On the other hand, Lenkei and colleagues [16] found that dogs went through an opening that was of similar height to their height at the withers without hesitation, but we decided that now this would be a too large of an opening for becoming the ‘small door’. Therefore, the smaller door height was always set as being shorter than the dog’s height at the withers by at least 11, maximum 20 cm. On the other hand, the large door was at least as tall as the dog’s height at the withers. As a summary, due to these measurements, the large door was convenient for the dogs, while in theory they would just fit through the small door.

### 2.3. General Procedure

Before the experiment, the owners were informed about the procedure while the dog was on a leash. The experimenter (E) measured the dog’s height at the withers while the dog stood still (the owner (O) was asked to assist when it was necessary). The measuring tape was held in a taut, vertical line while the measurement was taken. The E also took a picture of each dog’s head from above (perpendicular to the top of the skull), for subsequent cephalic index calculation.

We used treats or toys to motivate the dogs. Before the test, we asked the O what their favorite treat or toy was for motivating their dogs, and we used the specified treat/toy during the experiment. As a warm-up phase, at first the E gave a piece of treat to familiarize the dog with the food plate being used during the test. In the case of toys, the E initiated a short play session with the dog by using the toy. Most dogs were tested with a food reward, the toy was only the preferred option for four subjects.

The dogs were sorted into three testing groups. As much as possible, we balanced the subjects among the groups according to their sex, housing, and training conditions. Each dog was tested only in one group. Before each trial, the O positioned the dog on the starting point that was 2 m from the fence. This way, the longer route (detouring the fence) was approximately 7 m long, compared to the 2.1 m long shorter route (going through the opening).

#### 2.3.1. Groups Without Detour Demonstration (Large Door, N = 15; Small Door, N = 15)

Dogs were tested in 6 consecutive trials. Before each trial, the O positioned the dog at the starting point, 2 m from the fence. At the beginning of the first trial, the E asked the O to turn away with the dog, while she walked behind the short fence and signaled the O to turn back with the dog. The E showed the toy or the plate with a treat (‘target’) and loudly called the dog’s attention (by saying the dog’s name and the word “Look!”). Then, the E placed the target on the ground, 15 cm from the fence. The E stepped back 0.5 m, signaled to the O to release the dog and started the stopwatch. The O was allowed to encourage the dog to obtain the reward, but they were asked to stay at the starting point and not use any directional commands or cues (such as pointing with their hand).

Dogs had the first three trials with closed doors and then, depending on their group designation, three trials with either the small or the large door open. If the dog reached the target, or after 1 min elapsed, the trial was over, and the O called the dog back. When the E set the opening before trial 4, the O with the dog was asked to turn away, thus preventing the dog from looking at the fence.

#### 2.3.2. Group with Detour Demonstration (N = 15)

Dogs were tested in 6 consecutive trials. In the first three trials the doors were closed, then in the next three trials the ‘large door’ was open for them. At the beginning of each closed-door trial the O and the dog faced the fence, while the E stepped in front of the dog and started to detour the fence to place the reward behind it. While performing the demonstration, E kept calling the dog’s attention with ostensive signals (calling the dog’s name, and saying ‘Look’, ‘Here I go’, etc.). At the beginning of the detour, the E walked towards the free end of the short fence, then turned behind it and walked along the fence until she arrived at the door. There she conspicuously held up the reward, then placed it on the ground and showed her empty hands towards the dog. The E stepped back, then signaled the O to release the dog. After the dog obtained the reward, O called the dog back and turned away with it, so the dog could not see as E came back from behind the fence to demonstrate the detour before the next trial. After three demonstrations, the next three trials with the large door open were run like the groups with no detour demonstration.

### 2.4. Behavioral Variables

Table 3 shows the behavioral variables used for analysis. We used BORIS software (v. 7.13.6 © Olivier Friard and Marco Gamba) for the extraction of data from the video sequences. To check the reliability of the coding, an independent observer (who was blind to the test hypotheses) coded the video footage from nine randomly chosen dogs (20% of the videos). Based on the analysis, our coding procedure was reliable (Spearman’s rho—Reward latency: R(54) = 0.995; *p* < 0.001; ‘Direction of approach’ R(54) = 0.697; *p* < 0.001; ‘Gazing at owner’ frequency: R(54) = 1.000; *p* < 0.001; ‘Looking at door’ frequency: R(54) = 0.750; *p* < 0.001; ‘Demonstration watching in Trial 3’ duration: R(9) = 1.000; *p* < 0.001).

### 2.5. Cephalic Index Calculation

We measured the dog’s cephalic index from photographs taken at the beginning of the experiment. The cephalic index value was calculated from photographs with GIMP image editing program 2.2.13. (http://www.gimp.org/, accessed on 1 July 2024) using its ruler function that takes the measurements in pixels. The skull width was measured between the two zygomatic arches, while the length was measured between the top of the nose and the occipital protuberance. The cephalic index was calculated as the ratio of the width of the head divided by the head’s length, multiplied by 100.

### 2.6. Statistical Analyses

The statistical analyses were performed with SPSS.29 (IBM Inc., Chicago, IL, USA).

Whenever it was feasible, we added the biologically meaningful two-way interactions to the initial models (testing group with housing conditions, testing group with training level). Then, we applied backward model selection, by removing the non-significant interactions one-by-one, always starting with the one with the highest *p*-value. Everywhere we report the final (simplest) model’s result.

In the case of relative frequencies, durations, and latency-type data, we checked the normal distribution of our data by visual inspection of the Q-Q plots of residuals. Latencies were then analyzed with repeated measures General Linear Models (GLM), where the trial was the repeated factor, and depending on the given analysis, the testing group, type of demonstration (without vs. with detour demo) and door size (small vs. large), were the fixed factors. The relative frequencies and durations were analyzed with repeated measures GLMs, where we used the test group, keeping condition, and training level as fixed, and the trial as repeated factors. The ‘keeping condition’ and ‘training level’ have been added to the analysis of ‘gazing at the door’ and ‘gazing at the owner’ variables, because one could expect that these confounders may affect dogs’ interactions with their owners.

We used Generalized Estimating Equations (GEE) with binary logistics for analyzing the choice of solution and direction of approach. The dog’s ID was the random factor, the trial was the repeated factor, and depending on the analysis, dogs’ sex and testing groups, were added as fixed factors. Dogs’ sex was used in the case of the choice of solution, as sex can influence the size of dogs, thus in theory larger males could be more likely to opt for the detour than using the door.

The relative duration of watching the demonstration was analyzed in connection with reward latency (Pearson correlation); the choice of solution in trial 4 (unpaired t-test); and the door approach and approach direction in trials 1–3 (GEE with binary logistics).

The cephalic index values across the testing groups were compared with one-way Analysis of Variance (ANOVA). We analyzed the association between CI (as continuous variable) and choice of solution (Trials 4–6) with unpaired t-tests, and reward latencies with linear regression.

## 3. Results

In the closed-door trials (trials 1–3), the testing group (F(2,42) = 1.539; *p* = 0.226) and demonstration (F(1,43) = 0.160; *p* = 0.691) did not have a significant effect on reward latency. However, the repetition of trials influenced the latency (F(2,84) = 13.986; *p* < 0.001). Regardless of the group, the dogs reached the reward faster from trial to trial (Figure 3). When we examined whether the dogs became faster across the trials within the groups, we found significant results for all three groups (small/no detour demo: F(2,28) = 3.792; *p* = 0.035; large/no detour demo: F(2,28) = 3.660; *p* = 0.039; large/detour demo: F(2,28) = 6.293; *p* = 0.006). Detour latencies showed the most pronounced decrease between the trials in the group where the dogs observed the demonstration (Figure 3).

In the open-door trials (trials 4–6), the testing group (F(2,42) = 2.332; *p* = 0.110), detour demonstration (F(1,43) = 1.379; *p* = 0.247) and the repeated trials (F(2,86) = 0.200; *p* = 0.819) did not have a significant association with the reward latency; however, the door size did (door type: F(1,43) = 4.750; *p* = 0.035). Dogs in the small door without detour demo group were significantly slower than the dogs in the large door groups (Figure 4).

We also examined whether opening of the door, i.e., the transition between trials 3 and 4, affected the reward latencies. Trials (F(1,43) = 2.480; *p* = 0.123) and detour demonstration (F(1,43) = 2.024; *p* = 0.162) had no significant effect. However, we found that the dogs were significantly slower in the small-door group than in the other two groups (testing group: F(2,42) = 4.296; *p* = 0.020) and overall, the dogs that encountered the small door were significantly slower than the dogs that encountered the large door (door type: F(1,43) = 8.795; *p* = 0.005).

The dogs’ choice of solution for obtaining the reward had a significant association with the trial (Wald Chi^2^_(2)_ = 7.897; *p* = 0.019). Going through the door was more common in the last trial than in Trial 4, when the dogs encountered the open doors for the first time (Figure 5). The other fixed factors did not have significant associations with the choice of solution (sex: Wald Chi^2^_(1)_ = 0.016; *p* = 0.900; detour demonstration: Wald Chi^2^_(1)_ = 1.078; *p* = 0.299; door type Wald Chi^2^_(1)_ = 1.047; *p* = 0.306).

We analyzed whether watching the experimenter during the demonstration was associated with the dogs’ reward latencies (trials 1–3), door approach (trials 1–3), approach direction (trials 1–3), and choice of solution (in trial 4). We found no significant association with either the door approach (Wald Chi^2^_(1)_ = 0.234; *p* = 0.629), approach direction (Wald Chi^2^_(1)_ = 0.200; *p* = 0.655), trials (Wald Chi^2^_(2)_ = 1.296; *p* = 0.523), or reward latency (Pearson R(45) = −0.123; *p* = 0.421). However, the choice of solution in trial 4 had a significant association with the relative duration of watching the demonstration in the preceding trial (unpaired *t*-test, t(11.718) = −2.334; *p* = 0.019); according to this, those dogs continued to detour in trial 4, who spent a longer time looking at the demonstration in trial 3 (Figure 6).

In the case of the direction of approach, we examined whether the dogs started to move straight forward towards the door, or if they ran diagonally towards the free end of the fence. In the closed-door trials, the trial had a significant association with the approach direction (Wald Chi^2^_(2)_ = 6.880; *p* = 0.032) (Figure 7). Dogs were more likely to run straight ahead in Trial 1 and run towards the end of the fence in trials 2–3 (to make a detour). Trial also had a significant association with the direction of approach in the case of trials 3 and 4, when the door was open (Wald Chi^2^_(1)_ = 6.795; *p* = 0.009). Regardless of the size of the door (Wald Chi^2^_(1)_ = 0.416; *p* = 0.831), dogs tended to run towards it when this solution was available. In the open-door trials (trials 4–6), we did not find significant association with any of the repeated/fixed factors (trials: Wald Chi^2^_(2)_ = 1.810; *p* = 0.405; door size: Wald Chi^2^_(1)_ = 0.025; *p* = 0.874). When the door was open, dogs did not change their approach direction across the trials, they mostly ran straight towards the door.

With regard to looking at the owner, we found a significant trial effect (F(5,120) = 6.016; *p* < 0.001). From trial-to-trial, dogs looked at the owner less frequently, similarly in each group (F(2,24) = 0.369; *p* = 0.695) (Figure 8). Keeping condition (F(1,24) = 1.841; *p* = 0.187) and training level (F(5,24) = 0.369; *p* = 0.865) did not have significant associations with this parameter.

With regard to the frequency of looking at the door, we found strong trial effect (F(5,120) = 5.650; *p* < 0.001). The other fixed factors did not show significant association with this parameter (group: F(2,24) = 0.142; *p* = 0.869); keeping condition (F(1,24) = 0.090; *p* = 0.766) and training level (F(5,24) = 1.535; *p* = 0.217). We did not find significant interaction between the fixed factors. When the doors were open, independently from the testing group, the dogs looked at them more often than in the closed-door trials (Figure 9).

We assessed whether the dogs’ Cephalic Index values were the same across the test groups (Figure 10). We found no significant difference between them (F(2,39) = 0.297; *p* = 0.745). According to [42], our sample falls mainly within the mesocephalic category (CI 50–60), with a slight bias towards the shorter-headed dogs (above CI-60).

Additionally, we checked whether the dogs’ cephalic index had an association with their solution choice in the open-door trials (trial 4–6). We found no significant association in any of the trials (Trial 4: t(38) = −1.081; *p* = 0.286; Trial 5: t(38) = −0.644; *p* = 0.524; Trial 6: t(38) = 0.010; *p* = 0.992). We did not find significant association with the reward latencies either (Trial 1: Β = −0.592; SE = 0.517; t = −1.144; *p* = 0.260; Trial 4: Β = 0.099; SE = 0.055; t = 1.812; *p* = 0.078; Trial 6: Β = 0.118; SE = 0.087; t = 1.348; *p* = 0.185).

## 4. Discussion

In our study we found that when both solutions were available for the dogs to obtain the reward (i.e., the shortcut through the door and detour around the obstacle), dogs’ choices were mainly body awareness based: when the door was open, the reward latencies were no longer affected by the previously seen detour demonstrations. In this case, the most important factor was door size: dogs reached the reward sooner when they were provided with the large door compared to the small door group. The choice of solution in the case of open-door trials was mainly affected by trials: dogs increasingly chose the door as the trials went on. However, in the case of the small door, at first more than half of the dogs opted for the detour. This result underlines the importance of body awareness: dogs readily assessed the size of the opening during their first encounter with it and mostly opted for a detour in the case of the less comfortable (small) door. As the trials proceeded, dogs looked back at the owner less frequently, regardless of the group. Looking at the owner is a sign of dependency, typical for dogs who struggle with a task [40,43]. As the dogs opted for the ‘optimal solution’ (shortcut) more often, they solved the task more easily and faster, especially in the last three trials. Although social learning (about the ‘suboptimal’ solution—the detour) was found to have less effect in this scenario, it also contributed to the complexity of the dogs’ decision-making. First, observing the experimenter demonstrating the detour resulted in a markedly decreasing latency of reaching the reward in the closed-door trials. Interestingly, in the detour demonstration group, those dogs that spent the longest time watching the demonstrator in the last closed-door trial, were more likely to continue to detour in the first trial even if they faced the comfortably large open door (‘optimal shortcut’).

In this experiment, dogs encountered a spatial problem-solving task with more than one solution. We expected that they would rely on their body-size representation when they had to decide between a shortcut, or a longer detour around the obstacle. Our results showed that dogs rely on their body size more convincingly compared to what was previously found [27]. In our current experiment, even the small door was big enough for the dogs to reach the reward, but they still assessed the size of the door and showed signs of body awareness. When dogs could choose between both solutions (i.e., in the open-door trials) and they encountered the small opening, they solved the problem more slowly—either because they hesitated longer, or because they more often opted to detour instead of using the door. The latter explanation is plausible based on the choice of solution results in the small door group’s fourth trials: here alone, the ratio of detouring dogs was above 50%. This result shows that dogs assessed the size of the available opening in comparison to their own size. A similar hesitating reaction was found in the case of approaching uncomfortably small apertures in the case of dogs [16,27], cats [23]; and body-size awareness was detected in the form of selecting the aperture with the optimal dimensions (compared to the animal’s body) in the case of ferrets [44], and rats [45]. This effect was detectable in the transition phase of opening the door as well. The transition between trial 3 and 4 was the most crucial moment, because dogs had been learning to detour routinely after three repetitions of closed-door trials. Then, suddenly a new possibility arose, a shortcut to reach the reward. Dogs that encountered the small door reached the reward significantly slower than dogs who were facing the large door. While in the choice of solution we did not find a significant association with the door size, in the case of the small door, almost half of the dogs still opted to detour the fence, while they showed an immediate, marked preference towards the large open door. In comparison, Pongrácz et al. [34] found that dogs who had the opportunity to detour three times without access to a shortcut, later almost uniformly opted for continuing to make a detour instead of going through the openings on either side of a V-shaped obstacle. In that earlier experiment the openings were always 40 cm tall, which is a suitable door size for every dog. However, due to the V-shape arrangement of the obstacle, these openings might be harder to notice for an approaching dog (for comparative results regarding detours around a straight vs. angulated obstacle see [46]).

Based on the dogs’ preference for approaching the fence diagonally in the first three trials, we could see that most dogs chose to make a detour right from the starting point; in other words, they could quickly learn and understand that they had to detour to reach the reward. Yet, when the experimenter opened the door, the dogs started to run straight towards it, independent of the door size. Even if the dogs finally opted for a detour, some of them still ran, at first, towards the open door. Based on the higher gazing frequencies towards the open door in the last three trials compared to the closed-door trials, we can conclude that the dogs most probably noticed the change (i.e., that the door has been opened), and not only looked towards the reward. If they would simply look at the reward through the mesh in Trials 1–3, there would be no difference in gazing time between the trials, whether the door was open or not. The small and large doors received comparable attention from the dogs, and they both were viable options for making a shortcut, unlike in the earlier study of Pongrácz et al. [27], where one of the doors was too small for the dogs.

In addition to body-size awareness, the second important factor that we expected to influence on the spatial problem solving of dogs was interspecific social learning. Dogs heavily rely on behavioral templates that were demonstrated by humans to the extent of copying irrelevant behavioral sequences (‘over imitation’: [47,48]). The strong influence of human motor actions on dogs is thought to be part of their dependence on humans, which is a hallmark of dogs’ evolution since domestication [49,50]. We expected that when the dogs could observe a human demonstrator repeatedly showing them how to detour the fence, this would override their subsequent preference towards using the optimal shortcut through the large door. However, we did not find a strong effect of social learning. Dogs from the detour demonstration group used the door just as often as the dogs from the groups without detour demonstration. However, contrary to previous social learning scenarios where it was found that dogs prefer the longer route around a V-shaped fence after they observed the demonstrator multiple times [34], here the door was right in front of the dogs because of the straight fence design we used, compared to the V-shaped fence we previously utilized. This setup could make the door harder to ignore for the dogs (we found a uniformly elevated gazing frequency towards the open doors in each group), and the experimenter also stood behind it when it was open. Thus, the dogs might learn to pay attention to the demonstrator herself and consequently, they might notice the door even more easily.

Can we say then that the demonstration of the detour was completely ineffective in this experiment? In our current study, most of the dogs could easily detour the one-ended, straight fence. This is in stark contrast with the difficulty that dogs encounter when they had to detour a V-shaped obstacle [40,41], where a substantial proportion of the test subjects failed to reach the reward within 60 s. Here, dogs quickly learned to detour around the straight fence and were faster along the trials in each experimental group. However, the detour demonstration also had an effect as the latency of reaching the target while the doors were closed had the strongest decrease in the demonstration group. Even in the case of such an easy detour task, social learning was detectable.

We found that the longer the dogs watched the demonstration in Trial 3, the more likely they continued to detour in Trial 4, when the door was already open. This indicates that the demonstration indeed influenced the solution choice; however, social learning was connected to the dogs’ actual attention towards the experimenter. This result aligns with our recent findings in the case of learning how to detour a V-shaped fence, where the relative duration of watching the demonstration also had a positive effect on the dogs’ eventual success rate [41]. Earlier, paying keen attention to the finer details of the actions of a human demonstrator was found to be of key importance when dogs had to memorize the location of a hidden object based on a video playback as well [51].

Apart from studies on inherited diseases and longevity (e.g., [52]) of companion dogs, non-purebred dogs of the global north are rarely targeted in scientific papers. This is in stark contrast with the growing number of publications about the behavior of free-ranging village (or ‘pariah’) dogs that represent a specific ‘ecotype’ of the species (e.g., [53,54]). Purpose-bred breed-crosses (‘designer dogs’) are also receiving more attention recently [55]. Breed-related behavioral differences, based on either functional selection (e.g., [41]) or ancestry-based effects (e.g., [56]) were found to be influential in the case of several socio-cognitive traits in dogs [6]. Here, (following the description of Gácsi et al. [39]) we focused on mixed-breed companion dogs that were apparently (according to their owners and to our visual inspection) neither direct breed-crosses nor non-pedigreed purebreds. There are so far no publications about the potential effect of dog breeds on body awareness, and as our experimental design was already considerably complex, we opted for the testing of the heterogenous and non-purposefully bred mixed-breed dogs. Most stray dogs are mixed breeds [57], which, when placed in a shelter, are also the hardest to re-home with adopting families (e.g., [58]). Mixed-breed dogs had to be successful in the anthropogenic niche without receiving appreciation from humans based on their value in any specific task. We can expect that mixed-breed dogs in general were adapted to succeed on their own to a much larger extent than purebred dogs. This may hypothetically include higher levels of independence for finding optimal solutions. We expected that in our tests, mixed-breed dogs would provide a solid baseline of canine sociocognitive performance in this complex, partly social problem-solving scenario. Our results showed that regardless of the group, the dogs detoured more often in Trial 4 than in Trials 5 and 6, so at first, they kept relying on using the solution that previously worked well for them. Overall, mixed-breed dogs relied only to a moderate extent on social learning, but they readily opted for the best (non-socially learned) solution on their own to reach the reward by using their body awareness to do so. This might be a sign of higher levels of independence in mixed-breed dogs compared to some of the purebred dogs that were selected for cooperative work with humans, similarly to the result that was found earlier in an experiment with point following [39]. However, we also found that mixed-breed dogs were ready to learn from the human demonstrator—this is a result that, so far, was not highlighted regarding mixed breeds.

We had another potential confounder that we were controlling for in our sample: the head shape of the dogs. Earlier we found that shorter-headed dogs more likely opted initially for using the door while the longer-headed dogs preferred the detour [27], but with no extreme cephalic index measures, now we did not find any effect of the head shape. The relatively homogenous sample (regarding the subjects’ cephalic indices) could lessen the potential effect of head shape in preferring one solution over the other.

### Limitations

A limitation of this study was that the door sizes were not specifically adjustable for each individual dog. However, this design was still more appropriate for testing the body awareness of dogs, as we could also provide passable small openings for the subjects. Moreover, we could also include dogs from a wider size range than in previous studies [27], thus our results are more representative. Another limitation could be that we did not run a small door/detour demonstration group. We chose not to test dogs in this scenario, because based on the results of [34], we expected that if dogs receive repeated demonstrations of the detour, if they face an inconveniently small door, they will most likely opt for the detour. Therefore, we wanted to investigate the strongest contrast between the offered shortcut (the large door) and the suboptimal, but socially reinforced detour, and the results supported our expectation that even in this condition, we could detect a modest effect of social learning.

## 5. Conclusions

In a realistic, ecologically valid study, where dogs could choose from various solutions and could optionally also rely on the template of human-demonstrated behavior, we showed that dogs combined body awareness and socially gained information in a complex problem-solving task. Mixed-breed dogs adapted to survive on their own in the anthropogenic niche, and here they readily opted for the optimal solution to reward maximizing when the door was comfortably large enough for them. Social learning had a weak but noticeable modifying effect, but only if the dogs paid keen attention to the demonstrator. We believe that reasonably complex testing scenarios, even if multiple factors are influencing the subjects’ behavior, can serve as the ‘best practice’ for a biologically meaningful ethological approach.

In this study we were interested in mixed-breed dogs as they were not (or only to a negligible extent) affected by artificial selection, in contrast to purebred dogs. But as the evolutionary history of dog breeds can also affect their social learning capacity [6,40], we have already embarked towards the next step, in which cooperative and independent working dog breeds are being tested in the same experimental setup as we have used here. As dog breeds are also characterized with genetically (more-or-less) fixed body dimensions and size [59], we expect that this might influence their body awareness capacity as well.

## Figures and Tables

**Figure 1 animals-15-00432-f001:**
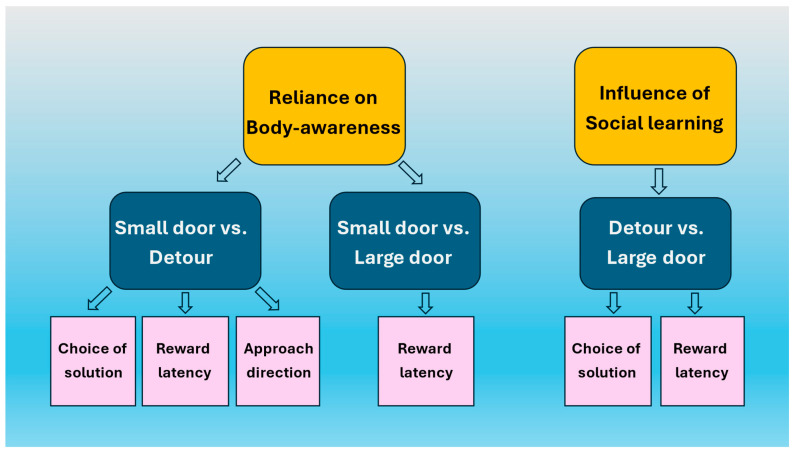
The flowchart of our experimental design. Yellow fields (top row): targeted phenomenon. Blue fields (middle row): dogs’ choices for comparison. Pink squares (bottom row): analyzed behavioral parameters.

**Figure 2 animals-15-00432-f002:**
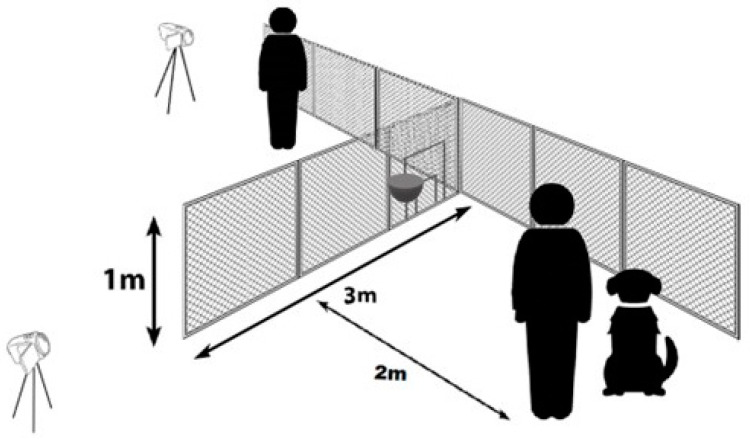
Schematic drawing of the experimental setup and cameras. The dog and owner are at the starting point. The experimenter stands on the other side of the fence. In the first segment of the short fence, the small and large doors are both visible in a closed position. The reward was placed on a plate on the other side of the fence.

**Figure 3 animals-15-00432-f003:**
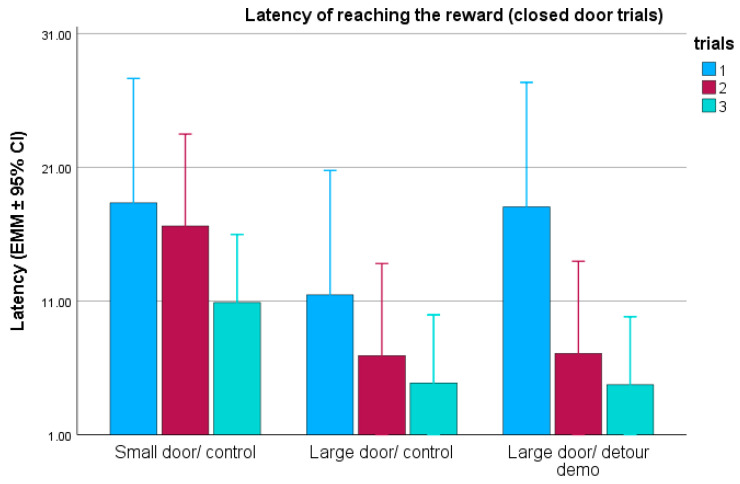
Reward latencies during the closed-door trials (trials 1–3), in the three testing groups. EMM = estimated marginal means; CI = confidence interval.

**Figure 4 animals-15-00432-f004:**
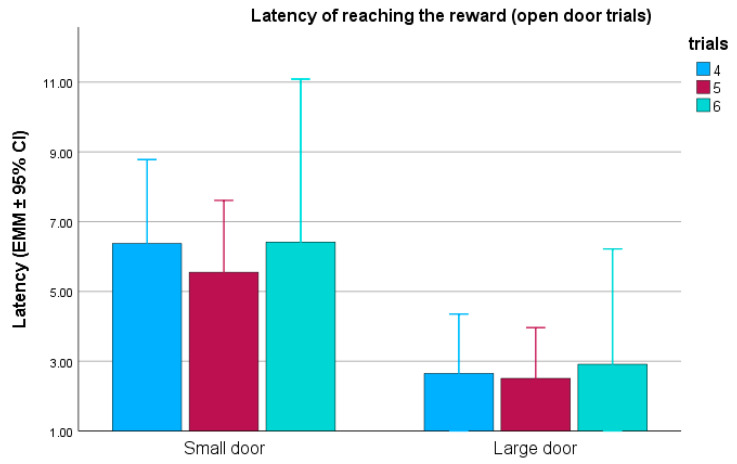
Reward latencies in the open-door trials (trials 4–6), according to the size of the doors. EMM = estimated marginal means; CI = confidence interval.

**Figure 5 animals-15-00432-f005:**
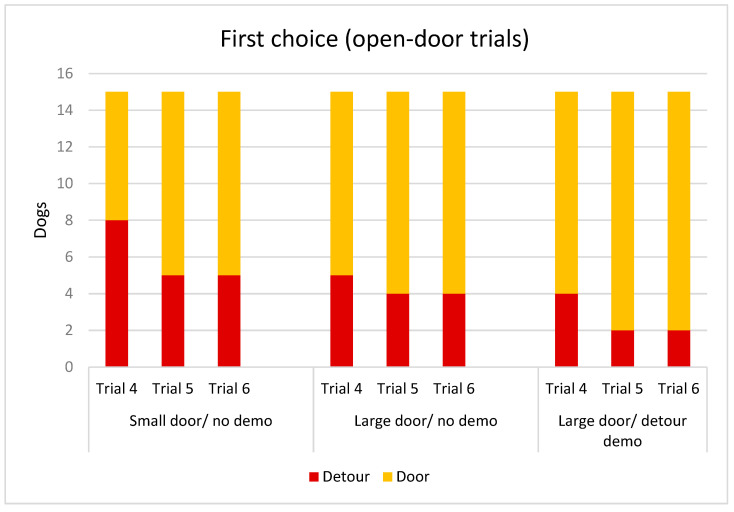
Number of the dogs who opted for the detour around the fence, or going through the doors in the last three trials in the different groups.

**Figure 6 animals-15-00432-f006:**
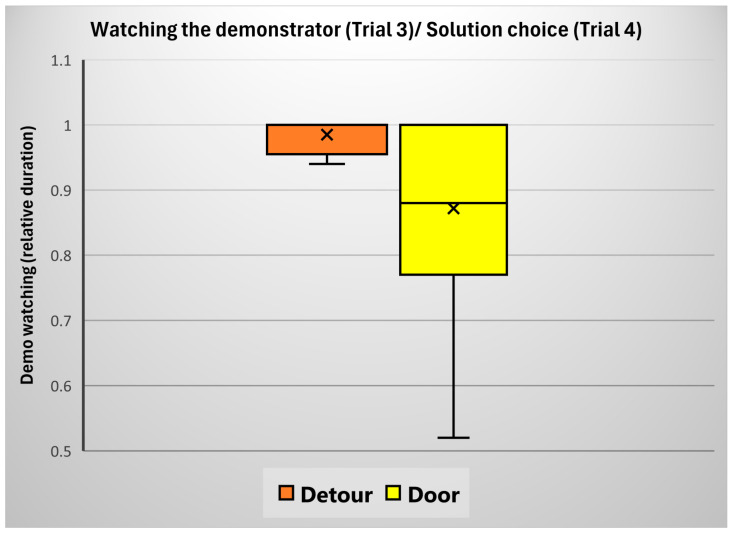
The relative duration of watching the demonstration in trial 3 in the case of dogs who either chose to perform a detour or go through the door in trial 4.

**Figure 7 animals-15-00432-f007:**
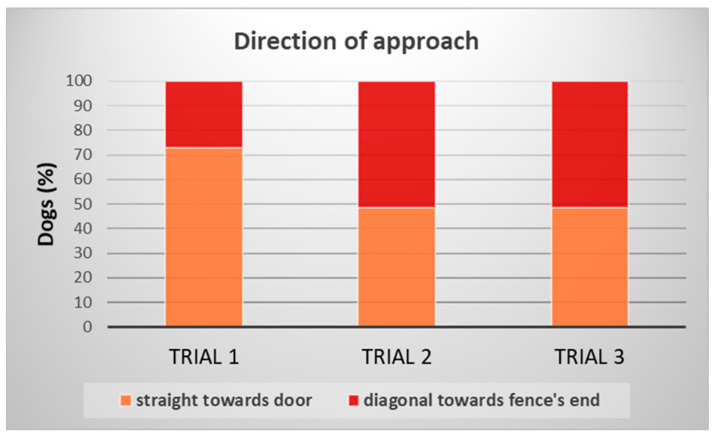
Proportion of dogs who chose one or the other direction when approaching the fence in the closed-door trials.

**Figure 8 animals-15-00432-f008:**
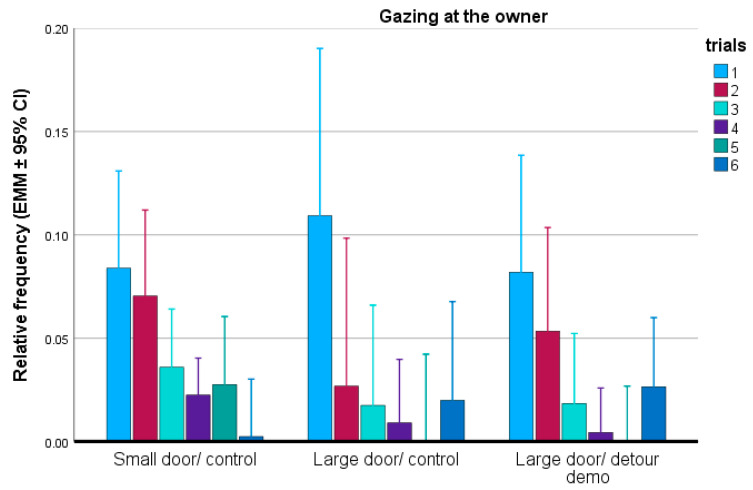
Relative frequency of looking at the owner in each group during the six trials. EMM = estimated marginal means; CI = confidence interval.

**Figure 9 animals-15-00432-f009:**
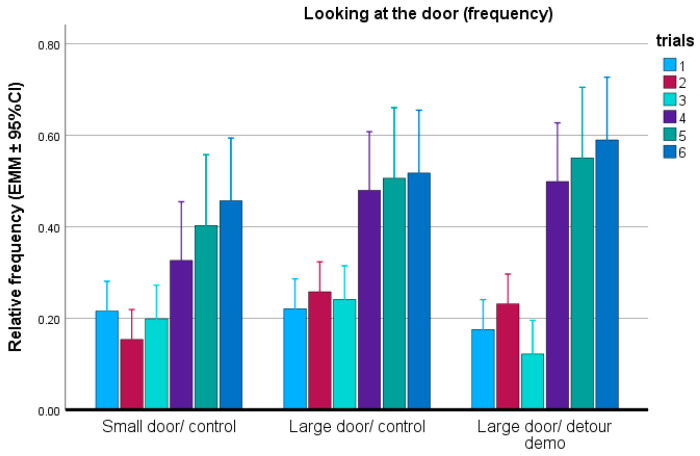
Relative frequency of looking at the door in each group during the six trials. EMM = estimated marginal means; CI = confidence interval.

**Figure 10 animals-15-00432-f010:**
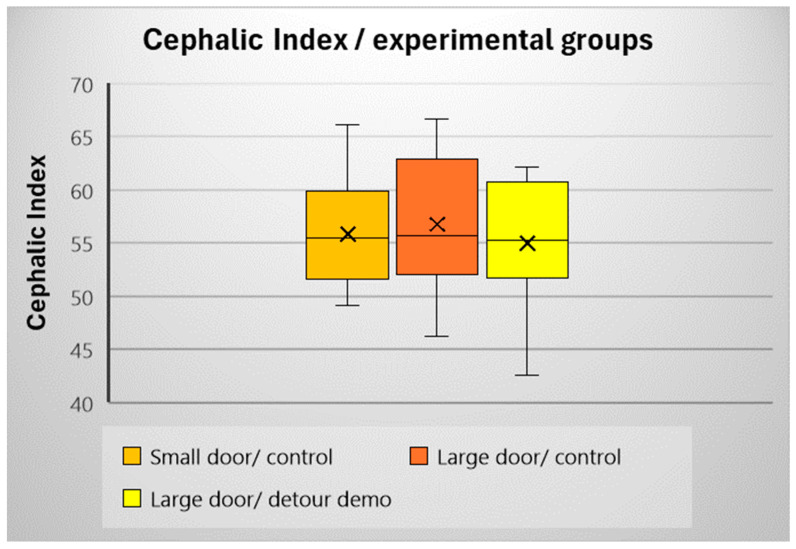
The cephalic index of dogs in each group (mean, median, upper, and lower quartiles).

**Table 1 animals-15-00432-t001:** Details of the dogs who participated in the experiment and their results which were included in the statistical analysis. L-nodemo = large door, without detour demonstration; l-demo = large door, with detour demonstration; s-nodemo = small door, without detour demonstration. NA = calculation of the cephalic index was not possible because of problems with the photograph of the dog’s head.

Dog ID	Test Group	Sex	Height (cm)	Cephalic Index	Age (Years)
1	l-nodemo	female	54	NA	12
2	l-nodemo	male	36	65.7	2.5
3	l-nodemo	male	59	55.3	7.5
4	l-nodemo	male	57	54.3	1
5	l-nodemo	male	64	62.5	14
6	l-nodemo	male	43	52.3	10
7	l-nodemo	male	54	47.8	3
8	l-nodemo	male	67	NA	5
9	l-nodemo	female	55	51.9	7
10	l-nodemo	male	57	59.3	1.5
11	l-nodemo	female	64	NA	5.5
12	l-nodemo	female	51	63	3.5
13	l-nodemo	female	51	66.6	7.5
14	l-nodemo	female	54	46.2	2
15	l-nodemo	male	57	56.1	5
16	l-demo	female	58	NA	14
17	l-demo	female	50	60.5	5
18	l-demo	male	60	61.2	13
19	l-demo	female	48	52.3	2.5
20	l-demo	male	65	55.2	4
21	l-demo	female	39	47.6	8
22	l-demo	female	52	61.7	5
23	l-demo	male	56	62.1	8
24	l-demo	male	33	52.7	13
25	l-demo	female	56	52.8	4
26	l-demo	male	54	55.3	6
27	l-demo	female	60	57.1	10.5
28	l-demo	female	43	58.9	2
29	l-demo	female	54	42.5	4.5
30	l-demo	female	68	49.9	2
31	s-nodemo	female	34	NA	6
32	s-nodemo	female	45	58.3	6
33	s-nodemo	male	58	53	12
34	s-nodemo	male	30	66.1	2.5
35	s-nodemo	female	53	49.1	11
36	s-nodemo	female	47	59.7	5
37	s-nodemo	male	40	58.1	3.5
38	s-nodemo	female	49	60.3	4.5
39	s-nodemo	male	59	52.9	4
40	s-nodemo	male	90	50.5	5
41	s-nodemo	male	43	58	5
42	s-nodemo	male	58	52.7	8.5
43	s-nodemo	female	45	60.3	3
44	s-nodemo	female	43	51.7	2.4
45	s-nodemo	female	58	51.1	6

**Table 2 animals-15-00432-t002:** The height of the doors in relation to the height of the dogs (measured at their withers). In parentheses, the Ns show the actual number of subjects in the given size category.

Dog’s Height (cm)	Small Door (cm)	Large Door (cm)
71–90 (N = 1)	70	-
61–70 (N = 5)	50	70
51–60 (N = 23)	40	70
41–50 (N = 10)	30	50
30–40 (N = 6)	20	40

**Table 3 animals-15-00432-t003:** The behavioral variables used for statistical analysis.

Behavioral Variable	Unit	Description
Success	Occurrence (0/1)	The dog touches/consumes the reward.
Choice (Trials 4–6 only)	Door/detour	The solution, used by the dog for reaching the reward.
Reward latency	(s)	Time elapsed between the moment of releasing the dog at the starting point and the dog’s arrival at the reward (i.e., after a successful detour or passing through the door).
Looking at the door	Relative duration	The dog turns its head towards the opening (total duration/reward latency)
Looking at the owner	1/s	The dog turns towards the owner (by turning its head only, or with full body orientation) and looks at them. The number of looking is divided by the reward latency.
Looking at the experimenter	1/s	The dog looks at the experimenter, who stands on the other side of the fence. The number of looking is divided by the reward latency.
Direction of approach	Straight or diagonal	The dog’s route after release. Straight means the dog run towards the door, diagonal means the dog started to run towards the free end of the fence.
Watching the demonstration (in Trials 1–3)	Relative duration	The dog looks at the demonstrator while she performs the detour (total duration of watching/demonstration duration).

## Data Availability

The original contributions presented in this study are included in the article/Appendix A. Further inquiries can be directed to the corresponding author.

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
