# Peer review of "Body Awareness Does Not Need a Pedigree: Mixed-Breed Dogs Rely More on Self-Representation Than Social Learning in a Spatial Task"

_animals, 2025, doi:10.3390/ani15030432_

Round 1

Reviewer 1 Report

Comments and Suggestions for Authors

Body awareness in dogs is an important element of their lives. It allows them to confidently move in their environment. As the authors mentioned several times, working dogs are focused on strong communication with their owner/handler. However, working dog breeds are often required to have a kind of "innate" body awareness, which helps in training such dogs and later helps the dogs in their work. Research on body awareness using mixed-breed dogs may allow for better understanding of this issue.

Comments:

The word "mongrel" commonly used by the authorsis not appropriate. It has a pejorative meaning, so a „mixed breed dogs” would be much better.

The study procedure is described quite clearly, but the names of the research groups are misleading, the control group is usually a group on which the experiment is not performed.

I assume that the name „Control group” was given because the Experimenter (E) did not perform the demonstration of walking around the fence, but this is not explained. The description of the „Demonstration group” is also not clear. Does this mean that the dog reached the reward bypassing the fence, or did it not have access to it before the trial with the open door? Because the description indicates the action of the Experimenter (E), and not the dog's:

line 253  - The E walked towards the free end of the short fence, then turned in and walked along the fence till she arrived at the door. The E conspicuously held up the reward, then she placed it to the ground and showed her empty hands towards the dog. The E stepped back, then signaled the O to release the dog. At the end of the trial, O called the dog back and turned away with it, so the dog could not see as E came back from behind the fence. After three demonstrations, the next three trials with the large door open were run like the control group.

The article “Body-Awareness Doesn't Need a Pedigree: Mongrel Dogs Rely on Self-Representation and Social Learning in a Spatial Task” meets the requirements for publication in the journal Animals and can be published after minor revision.

Author Response

Responses to REVIEWER # 1

Body awareness in dogs is an important element of their lives. It allows them to confidently move in their environment. As the authors mentioned several times, working dogs are focused on strong communication with their owner/handler. However, working dog breeds are often required to have a kind of "innate" body awareness, which helps in training such dogs and later helps the dogs in their work. Research on body awareness using mixed-breed dogs may allow for better understanding of this issue.

RESPONSE: We thank the Reviewer’s supportive opinion and the thoughtful comments. We adjusted the manuscript according to the suggestions. We provide in the following section detailed answers and explain how the text has been changed.

Comments:

The word "mongrel" commonly used by the authorsis not appropriate. It has a pejorative meaning, so a „mixed breed dogs” would be much better.

RESPONSE: Thank you for the advice, the term ‘mongrel’ was changed to ‘mixed breed dog’ everywhere, including the title and keywords as well.

The study procedure is described quite clearly, but the names of the research groups are misleading, the control group is usually a group on which the experiment is not performed. I assume that the name „Control group” was given because the Experimenter (E) did not perform the demonstration of walking around the fence, but this is not explained. The description of the „Demonstration group” is also not clear. Does this mean that the dog reached the reward bypassing the fence, or did it not have access to it before the trial with the open door? Because the description indicates the action of the Experimenter (E), and not the dog's:

RESPONSE: Thank you for this opportunity to clarify the methodology better. We renamed the groups, instead ‘Control’, we call the two groups where no detour demonstration happened ‘Groups without Detour Demonstration’, and the third group we call ‘Group with Detour Demonstration’. We also removed the ‘control’ word from the figures. We elaborated better the description of the methods, especially in the case of the group with detour demonstration. In each group dogs had 6 trials, where in the first 3 trials the doors were always closed, and in the last 3 trials either the small or the large door was always open (depending on the group, which one). Dogs could only obtain the reward if they made a detour in trials 1-3, while they had both options (detour or door) in the last 3 trials. In the case of the trials with detour demonstration (trial 1-3), the dog always observed the demonstrator’s action first, then it was allowed to obtain the reward by making a detour on its own. The amended text reads like this now (lines 373-396):

Dogs were tested in 6 consecutive trials. In the first three trials the doors were closed, then in the next three trials the ‘large door’ was open for them. At the beginning of each closed-door trial the O and the dog faced the fence, while the E stepped in front of the dog and started to detour the fence to place the reward behind it. While performing the demonstration, E kept calling the dog’s attention with ostensive signals (calling the dog’s name, and saying ‘Look’, ‘Here I go’, etc.). At the beginning of the detour, the E walked towards the free end of the short fence, then turned behind it and walked along the fence till she arrived at the door. There she conspicuously held up the reward, then placed it on the ground and showed her empty hands towards the dog. The E stepped back, then signaled the O to release the dog. After the dog obtained the reward, O called the dog back and turned away with it, so the dog could not see as E came back from behind the fence, to demonstrate the detour before the next trial. After three demonstrations, the next three trials with the large door open were run like the groups with no detour demonstration.

The article “Body-Awareness Doesn't Need a Pedigree: Mongrel Dogs Rely on Self-Representation and Social Learning in a Spatial Task” meets the requirements for publication in the journal Animals and can be published after minor revision.

RESPONSE: Thank you for the supportive comments, we appreciate them!

Reviewer 2 Report

Comments and Suggestions for Authors

The manuscript is interesting and original. It provides information about a controversial topic, scarcely studied in dogs. I have some comments that must be addressed before the manuscript can be considered for publication.

It is unclear to me how you analyzed the effects of the time spent watching the demonstration and the frequency of choosing the detour. In trial 4, how many dogs chose the detour? Was this number higher compared to that of whom chose the door? The proportion of dogs that preferred the detour strategy as a result of paying attention to the demonstration is unclear. I understand the association but I wonder how you conclude that this demonstrates the use of this strategy by de dogs.

In this regard, Figure 6 lacks clarity.

Based on that result, you made causal statements concluding that dogs use the human demonstration to solve the problem. However, in my opinion, the effect appears weak. Although the discussion briefly mentions the minor effect of the human demonstration, this is not consistent with the statements presented in the rest of the manuscript.

 I believe that the title is not accurate as it places effects of self-representation and human demonstration at the same level, even though animals showed they solve the problem mainly based on the size of the door.

You should provide a thorough justification for not including a small door-demonstration group considering this is a 2 x 2 design. In addition, that group could potentially reveal a more intense effect of the human demonstration.

Please state how you ruled out that the differences between small and large doors are not related to a salience effect rather than a body-size awareness phenomenon.  

The data analysis section is not clear. When and why did you include the effects of non-significant factors such as keeping, training and sex?. These factors were not included consistently along the analyses. In addition, you mention that the non-significant factors were removed from the final analyses.

Did you find any differences in dogs’ behaviors according to the selected reward (food vs toy)?. Sometimes each kind of reward can trigger different responses.

Is it possible that only 3 trials were not enough to learn the detour strategy?

L 311. “The strongest difference was found between the trials in the group where the dogs observed the demonstration” As far as I understand, you conducted a within-subject comparison, so this conclusion seems speculative.  

377… more often across the open trials or comparing close and open trials?

Please justify the statement “The small and large doors were equally interesting for the dogs” L 459

L 521 is too speculative; the current and previous data seems insufficient to draw this conclusion

Author Response

Responses to REVIEWER # 2

The manuscript is interesting and original. It provides information about a controversial topic, scarcely studied in dogs. I have some comments that must be addressed before the manuscript can be considered for publication.

RESPONSE: We highly appreciate your time and effort in providing us with valuable comments and suggestions. We are happy that the Reviewer found our research novel and interesting. We did our best to answer all the concerns and rewrite the text according to the suggestions. Our detailed responses can be seen below.

It is unclear to me how you analyzed the effects of the time spent watching the demonstration and the frequency of choosing the detour. In trial 4, how many dogs chose the detour? Was this number higher compared to that of whom chose the door? The proportion of dogs that preferred the detour strategy as a result of paying attention to the demonstration is unclear. I understand the association but I wonder how you conclude that this demonstrates the use of this strategy by de dogs.

RESPONSE: Thank you for the question. In the Detour Demonstration group, the doors were closed in Trials 1-3, and before each trial, the dogs watched the experimenter who demonstrated the detour to them. Dogs had only the detour as a viable option for obtaining the reward in the first three trials. However, in Trials 4-6, we opened the large door for them, so they could either detour or go through the door. In Trial 4 (the first such trial when they saw the open door), 4 dogs still opted for the detour and 11 used the large door. However, those dogs who chose to detour in Trial 4 watched the demonstrator in the previous trial significantly longer than those dogs who opted for using the door. This is why we interpreted this result as an indication of social learning: the longer the dogs watched the detour demo, the more likely they continued detouring even when the door was opened. We reworded the sentence about this result, we hope it conveys a clearer message now (lines 497-500):

choice of solution in trial 4 had a significant association with relative duration of watching the demonstration in the preceding trial (unpaired t-test, t(11.718)=-2.334; P=0.019); according to this, those dogs continued to detour in trial 4, who spent longer time looking at the demonstration in trial 3 (Figure 6).

In this regard, Figure 6 lacks clarity.

RESPONSE: We changed the title and the text on the vertical axis for better clarity, hopefully now Figure 6 is more understandable.

Based on that result, you made causal statements concluding that dogs use the human demonstration to solve the problem. However, in my opinion, the effect appears weak. Although the discussion briefly mentions the minor effect of the human demonstration, this is not consistent with the statements presented in the rest of the manuscript.

RESPONSE: We agree with the Reviewer that the effect of social learning was less relevant than the body-awareness effect. Thank you for your comment, we went through the text and toned down the interpretation of the social learning effect (Abstract, Discussion, Conclusion).

 I believe that the title is not accurate as it places effects of self-representation and human demonstration at the same level, even though animals showed they solve the problem mainly based on the size of the door.

RESPONSE: We took the Reviewer’s advice and rewrote the title, now it reads like this: “Body-Awareness Doesn't Need a Pedigree: Mixed Breed Dogs Rely More on Self-Representation than Social Learning in a Spatial Task”

You should provide a thorough justification for not including a small door-demonstration group considering this is a 2 x 2 design. In addition, that group could potentially reveal a more intense effect of the human demonstration.

RESPONSE: This is a valid question, thank you! We opted for running only the Detour demo/Large door open condition, because we wanted to investigate, whether dogs would show signs of social learning in the case of the ‘strongest contrast’ between the optimal shortcut (large door) and socially reinforced solution (suboptimal detour). The Reviewer is right, we also assumed that of we would couple the detour demo with the inconvenient small door, probably we could see stronger effect of social learning. This assumption was more-or-less tested in the earlier paper (Pongrácz et al., 2003), when it was found that dogs opted almost entirely for continuing the detours after the doors were opened on a V-shaped fence, and they witnessed demonstrations in the previous three (closed door) trials. In that experiment, the door was uniformly 40 cm tall, and the subjects were: Boxer (2), kuvasz (2), poodle (2), tervueren (2), beagle (1), malinois (1), mixed-breed (1), German pointer (1), Weimar pointer (1). The size of the mixed breed dog is unknown, but apart from it and the Beagle, the 40 cm door can be considered as a ‘small’ opening for the other subjects. Therefore, we can say that it was truly the Large door/ detour demo condition, which offered the most interesting combination for detecting the potential effect of social learning. Our results showed that dogs indeed used socially gained information in this case, although this effect was weaker than the solution based on body-awareness. We added some text about this issue to the Limitations of the study (lines 712-719):

Another limitation could be that we did not run a Small door/ detour demonstration group. We chose not to test dogs in this scenario because based on the results of [Pongrácz et al., 2003] we expected that if dogs receive repeated demonstrations of the detour, if they face the inconveniently small door, they will most likely opt for the detour. Therefore, we wanted to investigate the strongest contrast between the offered shortcut (Large door) and the suboptimal, but socially reinforced detour, and the results supported our expectation that even in this condition, we could detect a modest effect of social learning.

Please state how you ruled out that the differences between small and large doors are not related to a salience effect rather than a body-size awareness phenomenon.  

RESPONSE: Thank you for the relevant question. Truly, it is an important factor whether dogs equally noticed the small and large doors. We had two variables (‘looking at the door’, ‘approach’), which can be related to the salience of the doors. These variables did not show difference between the door sizes once the doors were open (see the results, approach: lines 511-518; gazing at the door: lines 530-535), thus we can assume that dogs noticed the small and the large door with comparable ease.

The data analysis section is not clear. When and why did you include the effects of non-significant factors such as keeping, training and sex?. These factors were not included consistently along the analyses. In addition, you mention that the non-significant factors were removed from the final analyses.

RESPONSE: Thank you for the questions. In the statistical analysis chapter, we described the method of model selection (lines 417-421), where we mentioned that the non-significant 2-way interactions were removed from the models:

“Whenever it was feasible, we added the biologically meaningful two-way interactions to the initial models (testing group with housing conditions, testing group with training level). Then, we applied backward model-selection, by removing one-by-one the non-significant interactions, always starting with the one with the highest p-value. Everywhere we report the final (simplest) model’s result.”

Thus, in the Results chapter, we always indicated the outcome of the analysis regarding each fixed factor, regardless of it was a significant or non-significant result. However, as we found no significant interactions between the fixed factors, these were all removed from the final models, and we did not report them. Keeping and training were included in those comparisons, where they could affect dogs’ gazing behavior due to the potential effects of these factors on dog-owner interactions. Dog’s sex was included to only one analysis, because based on the similar previous studies, we did not have otherwise a priori hypotheses about the effect of sex in this experiment, and the relatively low sample size warranted against ‘over parametrization’ of the analysis. However, we thought that as sex can affect dogs’ size (males are usually larger, at least in purebreds), we included this variable to the ‘solution choice’ analysis, where theoretically larger males could be expected to detour more often instead of using the opening (eventually sex did not have significant effect). We added more text to the description of our statistical analysis, to clarify these details (lines 432-434, and lines 438-440):

Keeping condition and training level have been added to the analysis of gazing at the door and owner variables, because one could expect that these confounders may affect dogs’ interactions with their owners.

Dogs’ sex was used in the case of choice of solution, as sex can influence the size of dogs, thus in theory larger males could more likely opt for the detour than using the door.

Did you find any differences in dogs’ behaviors according to the selected reward (food vs toy)?. Sometimes each kind of reward can trigger different responses.

RESPONSE: Thank you for this question. The task was not scent-based but mainly a visual one, thus we do not consider the using of one of the other types of reward as a relevant confounder. We allowed using food or toys as a reward to find the most motivating incentive for the dogs. Only 4 subjects were tested with a toy, all the others used food reward, thus any reward-dependent differences can be regarded as minimal. We added the number of toy-using subjects to the Methods.

Is it possible that only 3 trials were not enough to learn the detour strategy?

RESPONSE: This is an interesting question, perhaps more trials with closed doors and detour demonstration would result in higher preference for the detour strategy in the subjects. We opted for three trials here, because we wanted to conduct a comparable experiment to our previous papers (Pongrácz et al., 2003; 2023), where dogs had also 3 trials with closed doors, before they were offered with a shortcut.

L 311. “The strongest difference was found between the trials in the group where the dogs observed the demonstration” As far as I understand, you conducted a within-subject comparison, so this conclusion seems speculative.  

RESPONSE: Thank you for the comment. The Reviewer is right, this was a within-subject analysis here. We based our interpretation on the p-values in the three groups. While the significant effect was modest in the two groups, where dogs did not see demonstration (P=0.039 and P=0.035, respectively), in the detour demonstration group the effect of consecutive trials had a more pronounced significance (P=0.006). We changed the wording of this sentence (lines 455-457), now it reads like this:

Detour latencies showed the most pronounced decrease between the trials in the group where the dogs observed the demonstration (Figure 3).

377… more often across the open trials or comparing close and open trials?

RESPONSE: Thank you for the relevant comment. Here the effect of the group was not significant, and we did not find significant interaction between the trials and group either. The dogs looked more often at the doors when those were open, compared to the closed-door trials, independently of the group. We added some text for clarity (lines 533-535):

“We did not find significant interaction between the fixed factors. When the doors were open, independently from the testing group, the dogs looked at them more often than in the closed-door trials (Figure 9).”

Please justify the statement “The small and large doors were equally interesting for the dogs” L 459

RESPONSE: This section discusses the results of the previously mentioned gazing at the door behavior. We amended the text here for clarity, hopefully now it reads better (lines 616-624):

“Based on the higher gazing frequencies towards the open door in the last three trials compared to the closed-door trials, we can conclude that the dogs most probably noticed the change (i.e., that the door has been opened), and not only looked towards the reward. If they would simply look at the reward through the mesh in Trials 1-3, there would be no difference in gazing time between the trials whether the door was open or not. The small and large doors received comparable attention from the dogs, and they both were viable options for making a shortcut, unlike in the earlier study of Pongrácz et al. [27], where one of the doors was too small for the dogs.”

L 521 is too speculative; the current and previous data seems insufficient to draw this conclusion

RESPONSE: Thank you for the comment, this section truly needed some adjustment. Our goal is to draw comparison between our results (dogs relied less on social learning and more on body-awareness) and the earlier paper from Gácsi et al. (2009), who found that mixed breed dogs followed human pointing gestures les often than purebred dogs did. The new text reads like this (lines 692-698):

“Overall, mixed breed dogs relied on social learning only to a moderate extent, but they readily opted for the best (non-socially learned) solution on their own to reach the reward by using their body-awareness to do so. This might be the sign of higher levels of independence in mixed breed dogs compared to some of the purebred dogs that were selected for cooperative work with humans, similarly to the result that was found earlier in an experiment with point-following [39].”

Round 2

Reviewer 2 Report

Comments and Suggestions for Authors

I appreciate the modifications that have been made to the manuscript. In my opinion, the manuscript is suitable for publication.